Histological characterization of denticulate palatal plates in an Early Permian dissorophoid

Gee Bryan M. bryan.gee@mail.utoronto.ca
Haridy Yara
Reisz Robert R.
Department of Biology, University of Toronto Mississauga , Ontario , Canada
Marsicano Claudia
Electronic publication date: 2017 Aug 22
Publication date: 2017
Volume: 5
Electronic Location ID: e3727
Received 2017 Jun 21; Accepted 2017 Aug 2
Copyright: ©2017 Gee et al.
Copyright year: 2017
Copyright holder: Gee et al.
License: This is an open access article distributed under the terms of the Creative Commons Attribution License, which permits unrestricted use, distribution, reproduction and adaptation in any medium and for any purpose provided that it is properly attributed. For attribution, the original author(s), title, publication source (PeerJ) and either DOI or URL of the article must be cited.
License URL: https://creativecommons.org/licenses/by/4.0/

Keywords: Palatal plates, Denticles, Dissorophoid, Histology, Temnospondyl

Funding: Natural Sciences and Engineering Research Council (NSERC) of Canada Discovery Grant University of Toronto This work was supported by the Natural Sciences and Engineering Research Council (NSERC) of Canada Discovery Grant to RRR and the University of Toronto. The funders had no role in study design, data collection and analysis, decision to publish, or preparation of the manuscript.

==============================
Denticles are small, tooth-like protrusions that are commonly found on the palate of early tetrapods. Despite their widespread taxonomic occurrence and similar external morphology to marginal teeth, it has not been rigorously tested whether denticles are structurally homologous to true teeth with features such as a pulp cavity, dentine, and enamel, or if they are bony, tooth-like protrusions. Additionally, the denticles are known to occur not only on the palatal bones but also on a mosaic of small palatal plates that is thought to have covered the interpterygoid vacuities of temnospondyls through implantation in a soft tissue covering; however, these plates have never been examined beyond a simple description of their position and external morphology. Accordingly, we performed a histological analysis of these denticulate palatal plates in a dissorophoid temnospondyl in order to characterize their microanatomy and histology. The dentition on these palatal plates has been found to be homologous with true teeth on the basis of both external morphology and histological data through the identification of features such as enamel and a pulp cavity surrounded by dentine. In addition, patterns of tooth replacement and ankylosis support the hypothesis of structural homology between these tiny teeth on the palatal plates and the much larger marginal dentition. We also provide the first histological characterization of the palatal plates, including documentation of abundant Sharpey’s fibres that provide a direct line of evidence to support the hypothesis of soft tissue implantation. Finally, we conducted a survey of the literature to determine the taxonomic distribution of these plates within Temnospondyli, providing a broader context for the presence of palatal plates and illustrating the importance of maintaining consistency in nomenclature.

Introduction

Denticles are generally recognized as small, recurved tooth-like protrusions and are commonly found in high densities throughout the palatal region in Paleozoic sarcopterygians and tetrapods, including temnospondyls (Warren & Davey, 1992). They are also known from a subset of the more derived Meosozic temnospondyls, such as the Mastodonsauridae (e.g., Welles & Cosgriff, 1965), the Chigutisauridae (e.g., Damiani & Warren, 1996), the Plagiosauridae (e.g., Schoch & Witzmann, 2012), and the Rhytidosteidae (e.g.,Warren & Davey, 1992), supporting the hypothesis that the presence of denticles represents the plesiomorphic condition within Temnospondyli (Warren & Davey, 1992). Although denticles are frequently described in the literature, they are rarely defined, nor are they exclusive to early tetrapods. The term ‘denticle’ is also used in reference to serrations on the teeth of various vertebrate taxa, such as theropod dinosaurs (e.g., Smith, 2005; Brink et al., 2015) and other reptiles (e.g., De Andrade et al., 2010), the tooth-like scales on extant cartilaginous fishes (e.g., Serra-Pereira et al., 2008), and tooth-like protrusions in the pharyngeal cavity of agnathans, placoderms, osteichthyans, and basal tetrapods (e.g., Johanson & Smith, 2005; Witzmann, 2013). The term has also been applied to structures in non-vertebrate groups, such as the tooth-like protrusions found on the hinges of bivalves (e.g., Le Pennec, 1980) and on the radula in various mollusks (e.g., Lowenstam, 1962), the cheliceral teeth of sparassid spiders (e.g., Jäger, 1997), and the tooth-like processes on the subcapitulum of many mites (e.g., Evans & Till, 1979). It is even used to refer to early cytoskeletal components in Drosophila (e.g., Price et al., 2006).

In one of the few works to formally define anamniote tetrapod denticles, Lombard & Bolt (2003), revised an earlier definition of denticles by the same authors whereby denticles are considered to be any tooth-like protrusion possessing no more than 20% of the average maximum basal diameter and/or height of adjacent marginal teeth. However, this definition is only reflective of the external morphology, and it remains relatively untested whether denticles are true teeth or if they are simply a tooth-like protrusion formed from a different tissue and a different developmental process. For example, some clades of extant amphibians possess bony protrusions known as odontoid processes that superficially resemble enlarged teeth; these are found on the mandibles and are probably used in intraspecific competition (Fabrezi & Emerson, 2003). The only previous histological analysis of denticles in a temnospondyl is that of Bystrow (1938), who provided an exceptionally detailed description and drawings of histological sections of denticulate palatal bones of the Early Triassic trematosauroid Benthosuchus. Bystrow’s work provided clear and convincing evidence that the denticles in that taxon are structurally similar to the marginal dentition, but this work has been largely overlooked, perhaps due to being written in German, and further studies are needed to evaluate these findings. Denticles of extinct Paleozoic anamniotes are regularly found on the various bones of the palate, but there have also been a few documented occurrences of exceptional preservation of small denticulate palatal plates in temnospondyls that may have been attached to soft tissue coverings of the interpterygoid vacuities (e.g., Milner & Sequeira, 1998; Witzmann & Schoch, 2006 (for 2005); Schoch, 2006; Fröbisch & Reisz, 2008). These differ from denticulate branchial ossifications, which occur across a wide range of anamniote tetrapods (summarized in Witzmann, 2013), including colosteids (e.g., Hook, 1983; Witzmann, 2013), temnospondyls such as branchiosaurids (e.g., Boy, 1972), micromelerpetontids (e.g., Boy, 1995); dvinosaurs (e.g., Berman, 1973; Milner, 1982; Witzmann, 2004), eryopids (e.g., Boy, 1990; Witzmann, 2005), and various stereospondylomorphs (e.g., Schoch, 2002; Schoch, 2003; Schoch, 2008; Witzmann, 2006b; Damiani et al., 2009; Schoch & Witzmann, 2009), and the ‘microsaur’ Microbrachis (Olori, 2013). Based on comparisons of these ossifications in taxa in which both palatal and branchial plates are found (e.g., Onchiodon, Archegosaurus), they can be differentiated based on their position relative to other cranial elements and their general morphology; branchial plates are more elongate and oval in contour and feature only a single row of teeth at one edge (e.g., Boy, 1972; Witzmann, 2005; Witzmann, 2004; Witzmann, 2013). Some taxa (e.g., branchiosaurids, larval trematopids) feature isolated branchial denticles that are directly attached to the ceratobranchials, rather than on small plates (Witzmann, 2013), and were likely for filter feeding (Schoch, 2009). Additionally, the presence of branchial plates is indicative of the retention of gill arches with open gill clefts, whether in a larval form or in aquatic (paedomorphic and non-paedomorphic) adults, by virtue of their association with the branchial skeleton and should not occur in mature terrestrial anamniotes.

Palatal plates have not been previously examined in great detail and are only briefly discussed as an aside in broader descriptions by past workers. In any given publication, only a handful of references comprising a subset of other temnospondyl taxa in which these plates are known are provided, resulting in the perception that they are exceptionally rare. However, our review of the literature indicates that they are widely preserved in many temnospondyl families, and as discussed later in this paper, a diversity of nomenclatural terms used to refer to the plates likely plays a significant role in their misperceived paucity. Compared to other temnospondyl families, the plates are relatively well-represented within terrestrial dissorophoids, as they are known in the amphibamids Pasawioops (Fröbisch & Reisz, 2008), Platyrhinops (Carroll, 1964; Clack & Milner, 2010 (for 2009)), and Eoscopus (Daly, 1994), in the dissorophids Cacops morrisi (Reisz, Schoch & Anderson, 2009), Dissorophus sp. (BMG, pers. obs., 2017), Kamacops (Gubin, 1980), and Aspidosaurus binasser (Berman & Lucas, 2003), and in the trematopids Tambachia (Sumida, Berman & Martens, 1998), Fedexia (Berman et al., 2010), and Phonerpeton (Witzmann & Werneburg, 2017). A large number of palatal plates were also reported from “Broiliellus” hektotopos (Berman & Berman, 1975), originally described as a dissorophid, but the taxonomic affinities of this taxon are in need of revision, and it is not currently considered to be a member of the family in more recent phylogenetic analyses (e.g., Schoch, 2012).

The presence of denticle-bearing plates over the large vacuities would likely have served to significantly increase the overall surface area that could be used in prey capture. The paucity of these plates likely resulted from their soft tissue attachment and reduced preservation potential, rather than real anatomical absence, particularly in forms with extensive denticle fields on the palatal bones. It is possible that these plates are distributed among early tetrapods based on their documented presence in a colosteid-like form (Clack et al., 2012) and perhaps even in some derived sarcopterygians, as they are reported in the porolepiform Glyptolepis (Jarvik, 1972). However, because the plates show a strong correlation with the large interpterygoid vacuities that characterize temnospondyls, the denticulate palatal plates present in more basal taxa lacking such vacuities, such as those reported around the internal opening of the spiracle and on the basal plate of the parasphenoid in Eusthenopteron (Jarvik, 1954), may not be structurally or developmentally homologous to those of temnospondyls.

In this study, we are principally interested in the characterization of the small palatal plates and their denticles in early temnospondyls, with a focus on a specimen from the well-known Dolese Brothers Limestone Quarry near Richards Spur, Oklahoma. The superb quality of preservation of these small palatal plates and their dentition is unequalled in the fossil record, allowing us to study these plates in great detail. These palatal plates provide an excellent model system for studying the relationship between palatal and marginal dentition, and we propose to undertake their anatomical and histological study as part of a larger project on the evolution of palatal dentition. The purpose of this paper is threefold: (1) to histologically sample denticles of small palatal plates in order to re-test the original findings of Bystrow (1938) that indicated structural homology between denticles of the palatal bones with true teeth, (2) to histologically sample, for the first time, palatal plates of a dissorophoid that are comparable to those found across Temnospondyli, and (3) to discuss the phylogenetic origin and development of these plates. In addition to a description of the external morphology, we also incorporate histology, an informative technique for studying the internal structures and tissues. Although paleohistological studies of extinct fauna are becoming an increasingly frequent component of paleontological analyses, only a few have utilized amphibian teeth as the primary study structure (e.g., Bystrow, 1938; Schultze, 1969; Warren & Davey, 1992; Warren & Turner, 2005). This study provides an important follow-up to the work of Bystrow to test the hypothesis of structural homology between denticles and marginal teeth in temnospondyls. Finally, little attention has been directed toward the palatal plates of temnospondyls beyond a simple description, but recent work by various authors regarding the function of the interpterygoid vacuities and the histological work presented here permit an enhanced discussion of these ossifications. The characterization of denticulate palatal plates with acrodont implantation is important for improving our understanding of the dentition of temnospondyls, which has previously been restricted mostly to studies of the pleurodont marginal teeth (e.g., Warren & Davey, 1992). Denticulate palatal plates have never been histologically analyzed in early tetrapods; accordingly, the histological study of these structures fills a gap in the knowledge of early anamniote tetrapod dentition and their inferred feeding strategies.

Figure 1 The external morphology of the palatal plates in the sampled specimen (ROM 76838) and Pasawioops.

(A–C) images of the sampled block of palatal plates (ROM 76838) from various views; (D) an image of the dorsal surface of the palatal plates; (E) an enlarged view of the denticulate surface of the plate; (F) an individual tooth on the plate, showing fluting; (G) an SEM image of the block from which the plates were isolated; (H) enlarged SEM image of the denticulate surface of the plate; (I) an SEM image of a single tooth; (J) an image of the palatal view of the holotype of Pasawioops (OMNH 73019); (K) An enlarged view of the palatal plates; (L) an enlarged view of the dentition on the palatal plates showing the orientation of the dentition.

Materials & Methods

Materials

The sampled material consists of a small isolated block of palatal plates from an indeterminate dissorophoid temnospondyl (ROM 76838, Figs. 1A–1I). Plates are distributed on the top and the bottom of the block, as well as on the sides; some are articulated with adjacent plates while others are isolated. Some plates also overlie others, although this is not presumed to be the natural articulation condition. Most of the sampled plates are of the typical black coloration that results from hydrocarbon enrichment of the material during preservation at the Richards Spur locality, but several white, unenriched plates were also identified; these are of very low contrast to the lightly colored calcareous matrix and are only visible due to the presence of exposed pulp cavities infilled with a darker material. No other elements are present on the block. We make the taxonomic assignment to the Dissorophoidea on several lines of evidence: (1) among extinct anamniote tetrapods, denticulate plates are only known in temnospondyls, an observation that is likely phylogenetically correlated with the plesiomorphically large interpterygoid vacuities that characterize the clade and that are typically absent in lepospondyls, (2) of the few lepospondyls with convergently large interpterygoid vacuities (e.g., the nectridean Diplocaulus and the ‘microsaur’ Hyloplesion), none are reported to have palatal plates (e.g., Bossy & Milner, 1998; Carroll, 1998), and these taxa do not occur at the fossiliferous Early Permian karst deposit near Richards Spur, (3) terrestrial dissorophoids are the only temnospondyls known from the locality, and (4) the external morphology of the plates and the denticles on them are highly similar to those of the co-occurring amphibamid Pasawioops (Fröbisch & Reisz, 2008, Figs. 1J–1L) and the dissorophid Cacops (Reisz, Schoch & Anderson, 2009), in which the plates are articulated within the skull, and to those of the dissorophid Dissorophus in which they can be reasonably associated with the skull (BMG, pers. obs., 2017). The absence of aquatic forms and early larval stages of the dissorophoid taxa at the locality provides another line of evidence in support of their identification as palatal plates, rather than as branchial plates. The denticles of these plates feature striations, which are seen in the denticles of olsoniforms such as C. woehri and an indeterminate trematopid (cf. Acheloma) from Richards Spur (BMG, pers. obs., 2017; Fröbisch & Reisz, 2012), but not in those of Pasawioops (Fröbisch & Reisz, 2008), a pattern that also differentiates the marginal teeth of the olsoniforms from those of the amphibamids. Therefore, we propose that these plates either belong to a dissorophid or a trematopid, but because of the general paucity of the plates, a lack of knowledge of the relative size of the plates among the dissorophoid families, and a lack of knowledge of the ontogeny of the plates, we emphasize that the more specific identification is tentative and awaits further confirmation.

We also examined the holotype specimen of Pasawioops mayi (OMNH 73019) described by Fröbisch & Reisz (2008) that consists of a complete skull with articulated mandibles. More than two dozen semi-articulated palatal plates are found in the left interpterygoid vacuity and were briefly described and figured in the original description. Here we provide images of these plates at high magnification for comparison with those of ROM 76838 (Figs. 1J–1L). Although palatal plates are known from many temnospondyl taxa, this specimen represents one of the few documented occurrences of a nearly complete articulated mosaic preserved within the interpterygoid vacuity.

Histological analysis

ROM 76838 was imaged using a Leica DVM6 tilting microscope with LAS X software prior to sampling. Individual plates were removed from the block using an air scribe and a pin vise. A total of fifteen plates (all single elements save for an overlapping pair) were removed, divided between several containers, and glued to a pre-poured base layer of resin under a microscope so as to maximize the consistency of the orientation of the denticles for sampling. The plates were then embedded in a resin of Castolite AP and an associated hardener under vacuum and allowed to set for 24 h. The embedded plates were then separated into individual blocks that were hand-ground using both a Hillquist 1010 grinding cup and a lap wheel with a 600-mesh grit to remove the resin until a surface of the specimen was exposed. Each specimen was ground in a step-wise fashion to create a sagittal cut of each plate in the anteroposterior axis, with repeated examination under a microscope to evaluate the quality of the exposed cross-section. Because of the small size of the dental plates and corresponding challenges with their sectioning, the exposed plane differs slightly from an exact sagittal cut in some slides. Once an informative section was exposed, it was glued to a frosted plexiglass slide using Scotch-Weld SF-100 cyanoacrylate glue and allowed to set for at least 30 min. The slides were cut to a uniform height of 0.7 mm using a Buehler Isomet 1000 wafer blade low-speed saw and then ground on the Hillquist to achieve optical clarity. Slides were polished using a 1-micron grit aluminum oxide powder to remove polish lines without significant loss of material. Slides were imaged on a Nikon AZ-100 microscope, with a Nikon DS-Fi2 camera attachment, using Nikon NIS-Elements imaging software registered to RRR. Following the imaging of the initial slides of the indeterminate dissorophoid palatal plates, it was determined that cuts in a different plane were necessary to evaluate the possibility of plicidentine, and an additional six plates were isolated for the same preparation. Adobe Illustrator CS6 and Adobe Photoshop CS6 were used to compile figures and illustrations found within.

Description

External morphology

The external morphology of the denticles is similar to those seen on the denticulate palatal bones of temnospondyls (Figs. 1E–1F, 1H–1I). The denticles are conical at the base, as with the marginal dentition of many temnospondyls, with no clear compression in one axis compared to any other. Toward the crown, fluting on the teeth and the development of carinate edges become more pronounced. They are sharply recurved posteriorly, and the crowns, where preserved, are oriented nearly parallel to the ventral surface of the plate. It is important to note that patterns of external morphology of the teeth on the palatal plates likely mirror the marginal teeth. For example, the denticles of the holotype of Pasawioops (OMNH 73019), both on the palatal plates and on the palatal bones, are similar to the marginal teeth in that both bear no fluting and are not carinate (Figs. 1J–1L). This is in contrast to the dentition of the co-occurring olsoniforms, Cacops and Acheloma, both of which feature fluting and carinate edges on the marginal teeth and the denticles on the palatal bones (Reisz, Schoch & Anderson, 2009; Polley & Reisz, 2011). Vacant denticle sockets are often marked by a circular depression that is surrounded by an elevated ridge. There is no clear pattern to the distribution of vacant versus filled sockets, and due to the small size of the denticles, it is likely that some were accidentally removed during preservation or during preparation. Given that the denticles are significantly more numerous than the marginal dentition, they may not have followed a similar alternating replacement pattern. The arrangement of the denticles consists of even spacing and a relatively linear orientation parallel to the anteroposterior axis.

The plates bearing the denticles are smooth on the dorsal surface (embedded in the oral mucosa) and aside from the denticle sockets on the ventral surface, lack ornamentation on both dorsal and ventral surfaces. Many feature a slightly raised lip around the margin of the entire plate. The morphology of the plates is somewhat variable, ranging from quadrilateral to trilateral contours; the significance of this variation is unclear but does not appear to be the result of post-mortem damage; broken plates are clearly identifiable by the absence of a raised lip at a margin. Some plates clearly articulate with adjacent plates while others appear more isolated. This variation in plate morphology is also seen in other dissorophoids, such as Pasawioops (Fröbisch & Reisz, 2008), Platyrhinops (Carroll, 1964; Clack & Milner, 2010 (for 2009)), and Cacops (Reisz, Schoch & Anderson, 2009). In the plates sampled here, there is relatively little difference in size of plates. In contrast, the articulated plates of Pasawioops vary markedly in size; some are nearly twice as large as others, and many of those around the lateral margins of the vacuities are more circular and significantly smaller than those located within the interior of the vacuities. There is no apparent pattern of size distribution of the more interior plates. Additionally, some plates of Pasawioops, although relatively large, are nearly entirely smooth, with only a handful of denticles concentrated near the center. No such plates were present in ROM 76838. Finally, the holotype of Pasawioops features slender, elongate denticulate plates that overlie the cultriform process; the distinct morphology of these likely relates to a presumed attachment to the process rather than to a mucosa as with the plates that we sampled here.

Histological features

The denticles display essentially all of the main features that are considered to define teeth and the associated peridontia in that they are characterized by the presence of a pulp cavity, vascular canals, enamel, and dentine, complete with dentinal tubules and lines of von Ebner (Figs. 2 and 3). Slight variation in some features, such as the extent of the canals and the shape of the pulp cavity likely results from natural variation as well as the minor deviation from an exact sagittal cut, as noted in the methods (Fig. 3). As is further discussed below, comparisons of the various dental features between plates, such as enamel thickness, are not possible because of the slight variation in sectioning planes.

A pulp cavity is present in all of the sampled plates (Figs. 2 and 3). The exact morphology is somewhat variable due to the variation in sectioning plane. Sections in which the pulp cavity appears in-filled likely represent a section that captured the innermost dentinal wall that encloses the cavities. In some sections, an associated vascular canal(s) can be seen descending from the pulp cavity in a near-vertical orientation perpendicular to the dorsal surface of the plate (Figs. 2 and 3). The vascular canals are likely continuous with the dorsal surface of the plates; as with the pulp cavity, variation in the morphology and degree of penetration of the canals is likely the product of the plane of sectioning, as well as the three-dimensional nature of the canals. These canals are the only vascularity that can be identified in the sampled plates.

Figure 2 Histological characterization of two denticulate plates considered to be representative of the sample (ROM 76838).

(A–D) TS01135; (E–G) TS01140. (A) A schematic representation of TS01135 to show relative location of the histological features; (B) enlarged view of the crown of the tooth; (C) enlarged view of a fragment of dentine belonging to a previous generation of teeth at this position; (D) enlarged view of the Sharpey’s fibers that are embedded with in the plate; (E) enlarged view showing possible plicidentine at the junction between the tooth and plate; (F) enlarged view Sharpey’s fibers imaged under cross-polarized light; (G) the same view as (E) in cross-polarized light. Scale bar = 500 µm.

Figure 3 Histological sections showing incomplete ankylosis of the teeth to the palatal plates of ROM 76838.

(A) TS01142 showing incomplete ankylosis of a tooth to the plate; (B) close-up of the attachment site of the same tooth; (C) TS 01134 showing incomplete ankylosis of a tooth to the plate; (D) close-up of the attachment site of the same tooth; (E–F) incomplete ankylosis in TS01145 and TS01167. Scale bar = 500 µm.

Enamel is present on the majority of the denticles sectioned and is best visualized under cross-polarized light (Fig. 2G), although it can also be identified under plane-polarized light (Fig. 2A) and with the use of a lambda filter (Fig. S1A). The enamel is acellular and highly mineralized, covering the distal portion of the crown, and tapering towards but not reaching the attachment site. Measuring enamel thickness at the apex of the crown for comparable purposes between plates was not possible due to the aforementioned variation in sectioning.

Dentine is readily found in all of the sectioned dental plates, with pervasive dentine tubules found throughout the crown portion of the tooth, originating in the DEJ (dentine-enamel junction) and terminating in the pulp cavity (Figs. 2 and 3). The lines of von Ebner are readily identifiable in the mineralized dentine; unfortunately, in polyphyodont taxa, these lines are not informative to the age of the tooth (Erickson, 1996). The dentine-enamel junction is only identifiable in some sections (e.g., Figs. 2A and 2F, Fig. S1A).

Alveolar bone is easily identifiable as the attachment tissue, as it surrounds the basal edge of each tooth and is distinctly separated by a reversal line from the organized lamellar bone that makes up the plates (Figs. 2 and 3). Alveolar bone is typically fibrous or trabecular in appearance (sensu LeBlanc & Reisz, 2015) at the junction of the tooth and lamellar bone. However, it has a more paralleled-fiber appearance at the base of each denticle where it forms the base of attachment between the dentition and the plates, and post-mortem fractures tend to occur along this junction. The extent of ankylosis varies between sections and is often incomplete (Fig. 4). This cannot be attributed to a taphonomic loss of alveolar bone, as the unattached edge of the tooth is undamaged, while the opposing side is ankylosed to the plate by alveolar bone. There does not seem to be any pattern regarding the position of incomplete ankylosis; this can again be attributed to the variable planes of section.

Figure 4 Histological sections showing the variation in palatal plate anatomy of ROM 76838.

Scale bar = 500 µm. (A) TS01115; (B) TS01136; (C) TS01137; (D) TS01142; (E) TS01134; (F) TS01168; (G) TS00141; (H) TS01147.

Plicidentine, defined as infoldings of the dentine into the pulp cavity at the base of the tooth, has long been known in early tetrapods, being first documented in the temnospondyl Mastodonsaurus by Owen (1841). It has since been found to be widespread throughout numerous temnospondyls (often referred to as ‘labyrinthine infoldings’ in older literature) but is also known in a variety of extinct and extant fishes (e.g., Schultze, 1969; Schultze, 1970; Long, 1989; Meunier et al., 2015). Although the classical model of dental evolution suggested a loss of plicidentine in amniotes (e.g., Laurin & Reisz, 1995), these infoldings occur in a broad number of amniote groups, including ichthyosaurs (Maxwell, Caldwell & Lamoureux, 2011b), choristoderes (Gao & Fox, 1998), lepidosaurs (e.g., Kearney, Rieppel & Wood, 2006; Maxwell et al., 2011), captorhinids (De Ricqlès & Bolt, 1983), parareptiles (e.g., Modesto & Reisz, 2008; MacDougall, LeBlanc & Reisz, 2014; MacDougall, Modesto & Reisz, 2016), and synapsids (Brink, LeBlanc & Reisz, 2014). In lateral sections (Fig. 2E) of the plates and associated denticles, slight infolding of the dentine can be seen at the base of the tooth, dorsal to the pulp cavity, indicative of plicidentine. Although plicidentine is most easily visualized in a cross-sectional profile, this proved to be a particularly difficult section to produce, and our cross-sections in the dorsoventral axis could not produce clear evidence of plicidentine. We have tentatively identified weak infolding in the lateral profile (Fig. 2E), but additional work is required to verify the presence of this infolding.

In a single sectioned plate (Fig. 2B), a fragment of dentine can be seen embedded within the alveolar bone, neighbouring a complete tooth. This dentine fragment could have originated from a broken tooth; however, it should be noted that there are no other features to indicate the presence of a tooth that may have occupied this position, such as vascular canals or additional alveolar bone. The alveolar bone does not extend beyond the fragment of dentine, indicating that if alveolar bone had extended past the area where the fragment is embedded, it has been either eroded away, or remodelling has taken place. This dentine fragment likely indicates a position of a tooth from a previous generation that has since been replaced. This fragment, coupled with the varying degrees of ankylosis found in the dentition (Fig. 4), is indicative of cycles of tooth development and replacement.

The palatal plates are composed of compact lamellar bone that lacks many features commonly associated with bone, such as a diploë region, Haversian systems, and secondary bone. Present in all of the plates are incremental growth marks in the lamellar bone, which are identifiable by means of a variation in color associated with differential staining. The growth marks do not coincide with a change in density in the bone cell lacunae, which are densely distributed throughout the bone and which lack a consistent pattern in their arrangement. The osteocyte lacunae are consistently oblong in outline in all specimens; they are more numerous in some plates and often found in higher density at the tooth bases (Fig. 1E) or at the distal edges of the body of the plate (Fig. 4A, Fig. S1D). The canaliculi cannot be visualized, although this is likely the result of the relatively small scale of the plates. Primary or secondary osteons do not appear to be present in any of the sectioned plates, likely indicating slow sequential deposition of bone matrix. The absence of erosion cavities, as well as reversal lines (beyond those found at the tooth bases), also supports the characterization of the plates as lacking secondary remodelling. Also present in most of the palatal plates are abundant Sharpey’s fibers, which were identified on the basis of the large number of parallel striations that are oriented obliquely (mainly in the dorsoventral axis) to the body of the plate (Figs. 2D, 2F). The Sharpey’s fibers cut across the growth marks found in the bone and are present throughout the length of the plate, indicating that these plates were embedded in soft tissue throughout their development and subsequent growth. The Sharpey’s fibers appear relatively thin, but in some plates, there are high-density clusters of these fibers toward the curved ends of the plate. The significance of differences in the relative abundance and distribution of these fibers is not apparent in the absence of a positional context. Plates located at the margins of the vacuities where they abut against and possibly overlap onto the palatal bones may differ slightly in morphology compared to plates located in the interior of the vacuities that are only in contact with other palatal plates.

Discussion

Structural characterization of denticles

One of the main findings of this study is the characterization of the denticles of the palatal plates of a dissorophoid temnospondyl as being structurally identical to true teeth based on the presence of features such as enamel, dentine, a pulp cavity, and alveolar bone. The same features were identified in the denticles of the palatal bones of Benthosuchus by Bystrow (1938:figs. 26–27), supporting the hypothesis that denticles of the palatal bones and those of the palatal plates are structurally homologous to each other and to true teeth, although superficial aspects of the external morphology (e.g., striations, recurvature) are somewhat variable among temnospondyls, as with the marginal dentition. It is important to note that Bystrow himself paralleled the palatal denticles to dermal denticles of crossopterygian fishes, following Gross (1935), but he did identify the same typical dental tissues that we identify here in support of structural similarities. An additional consideration is the definition of ‘true teeth.’ Reif (1982:291) previously suggested that denticles should be differentiated from true teeth on the basis of their superficial formation in the mesenchyme, in contrast to the deeply invaginated dental lamina of ‘true teeth,’ such as the marginal dentition, in which replacement teeth form prior to the loss of the older teeth. However, Huysseune, Sire & Witten (2009:469) note that a dental lamina is not essential for tooth formation, thereby negating Reif’s distinction on the basis of the dental lamina and supporting the homology proposed in this study.

The presence of abundant Sharpey’s fibers provides histological evidence to support the hypothesis that the plates were embedded in the soft tissue membrane covering the interpterygoid vacuities. The possible identification of plicidentine may be useful for providing new insights into whether the infolding is the result of phylogenetic inheritance (e.g., Schultze, 1969; Schultze, 1970; Hill, 2005), functional significance in reinforcing the tooth attachment (e.g., Scanlon & Lee, 2002; MacDougall, LeBlanc & Reisz, 2014), or a combination of the two (e.g., Maxwell, Caldwell & Lamoureux, 2011a). However, additional sampling is necessary to determine with greater confidence the presence of plicidentine and to evaluate the degree of infolding, if present. Variable degrees of infolding between palatal denticles (either on plates or on palatal bones) and marginal dentition could be reflective of the different modes of attachment (acrodont versus pleurodont) or the presumed differences in modes of replacement that result from variable attachment. As we sampled a dissorophoid taxon that is ecologically, phylogenetically, and temporally distinct from Benthosuchus and analyzed denticles of the palatal plates rather than those of the palatal bones, we believe there is good reason to infer that the structural homology of the marginal dentition with denticles of both palatal bones and palatal plates is widespread throughout Temnospondyli.

Function

The function of the palatal plates in temnospondyls is likely strongly correlated with at least some of the hypothesized function(s) of the interpterygoid vacuities. Several recent publications have suggested a multiplicity of functions of the characteristic vacuities, none of which are mutually exclusive, including: (1) a palatal buccal pump (Schoch, 2014), (2) additional muscle attachment sites and increased optimization of bite force and stress distribution forces (Lautenschlager, Witzmann & Werneburg, 2016), and (3) retraction of the eyeballs during feeding and accommodation of cranial musculature (Witzmann & Werneburg, 2017). The retraction of the eyeballs in a manner similar to some extant lissamphibians to facilitate swallowing of large prey items (e.g., Deban & Wake, 2000; Levine, Monroy & Brainerd, 2004) is particularly appealing in the context of this paper, as the presence of a denticulate, flexible membrane would likely facilitate this process. Such flexibility could only be maintained through a sheet comprised of relatively small plates regardless of the overall size of the skull (Fig. 5); in all taxa documented to have these plates by our survey of the literature, the plates remain relatively small and likely increase primarily in count, rather than in size, in larger taxa (Fig. 6). Independent of the mobility of the vacuities, the strong recurvature of the teeth and their dense concentration, forming a continuous sheet with the palatal bones, would likely have facilitated increased contact with prey items and a unidirectional movement of prey toward the throat in concert with the tongue. Other proposed functions have less direct bearing on the function of the plates, although modeling approaches similar to those employed by Lautenschlager, Witzmann & Werneburg (2016) that incorporate the presence of the denticulate sheet formed by the plates could be helpful in assessing whether the plates influence bite force and stress distribution mechanics by providing a semi-rigid covering within the vacuity.

Figure 5 A schematic diagram featuring a cross section of a dissorophid skull with the position of the denticulate palatal plates in the interpterygoid vacuities.

(A) representation of the palatal plates in the oral mucosa covering the interpterygoid vacuity in a resting state; (B) representation of the plates during ventral expansion of the epithelium (e.g., during feeding).

Development

Although the use of extant analogues for inferring patterns of evolution in extinct relatives has often proven useful in anamniote tetrapods (e.g., Schoch, 2009; Fröbisch et al., 2010), no modern analogue for palatal plates embedded in soft tissue exists in any extant terrestrial tetrapods, let alone in any lissamphibians. As a result, the developmental origin and trajectory (e.g., replacement of worn or damaged denticles and plates, response of the plate mosaic to the expanding vacuity during ontogeny) of the plates remains somewhat unclear. It is impossible to correlate these plates with any ontogenetic stage given their isolation and the lack of contextual information from other taxa. Even in taxa like Pasawioops (Fröbisch & Reisz, 2008), with plates that are confidently associated with a skull, the plates are rarely known from more than one specimen in which the ontogenetic maturity of the skull is constrained only relatively, rather than absolutely. The most parsimonious explanation at present is that each plate represents an individual ossification, originating within the covering of the interpterygoid vacuity. This could account for the diversity of morphotypes seen among the plates of any one individual; naturally variable rates of expansion of the plates would lead some to be slightly larger that others, and also for some to acquire different shapes based on constraints imposed by surrounding plates.

Replacement

The replacement of the pleurodont marginal dentition found in dissorophoid temnospondyls is most likely by a single tooth position, although it bears noting that this is often assumed due to the relative homogeneity of tooth attachment in temnospondyls and extant lissamphibians and has not been rigorously tested in extinct anamniote tetrapods (Davit-Béal et al., 2007). The replacement of denticles is certainly difficult to define, considering that their attachment is acrodont. While acrodonty is frequently associated with a lack of replacement in extant reptiles, there is little work on the replacement of acrodont dentition of anamniote tetrapods (e.g., Bolt & DeMar, 1983), Here we present the most likely mode of replacement, where the dentition found on the palatal plates mirrors that of the pleurodont marginal dentition, in that a new tooth would form in soft tissue, ventral to the surface of the palatal plate and would eventually ankylose to the plate via alveolar bone. This process of progressive ankylosis was captured in several plates where the teeth were not fully ankylosed, leaving unattached edges of dentine (Fig. 4). Other teeth feature partial or ‘weak’ ankylosis, possibly indicating the progression of attachment. Similar incomplete ankylosis in denticles of Benthosuchus was figured by Bystrow (1938: figs. 26–27). It is important to note that this non-symmetrical ankylosis is not an indicator of hinged tooth attachment, as seen in snakes and some squamates (e.g., Patchell & Shine, 1986; Budney, Caldwell & Albino, 2006). This possibility can be ruled out on the basis that we observed fully ankylosed teeth; coupled with the absence of Sharpey’s fibers that would otherwise indicate the presence of the ligament necessary for hinged tooth attachment (Budney, Caldwell & Albino, 2006), this leads us to conclude that the area lacking complete ankylosis is related to replacement of the dentition. A new tooth would migrate towards an existing tooth position and eventually ankylose to the plate by eroding the existing tooth, leaving only a dentine fragment such as that identified in one plate (Fig. 2). However, an intermediate stage of the step-wise erosion was not captured histologically, by SEM, or with traditional imagining techniques in any of the plates. The same mode of tooth attachment is suggested by Bystrow (1938), but it is important to note that the replacement of denticles on the palatal bones is also characterized by successive deposition of new bone on top of existing denticles, which is not seen in the plates.

It cannot be excluded that the entire plate could be replaced if it was lost due to unnatural trauma, such as during feeding. Based on the apparent number of plates that would be expected in an individual such as the holotype of Pasawioops, the brief vacancy created by a shed plate would be unlikely to significantly impact the feeding success of the animal. There is no evidence to suggest that the plates would be regularly replaced as part of the normal ontogenetic trajectory, although this hypothesis cannot be excluded at present. However, it is unclear what characteristics would be useful for identifying shed or newly formed plates given the homogeneity and presumed comparable maturity in the sampled plates. Although the presence of variably sized plates could be interpreted as evidence of a natural replacement of palatal plates (assuming a positive relationship between size and maturity), other hypotheses cannot be discarded. For example, the small size and circular shape of plates around the margins of the vacuities in Pasawioops, a well-articulated occurrence of the plates, may be related to the articulation of these plates with the palatal bones. In the plates sampled here, there was very little variation in size; due to the semi-disarticulated nature of the material, it cannot be excluded that smaller plates were hydrodynamically sorted and removed from this sample. Overall thickness of the plate and the nature of the growth marks are also consistent throughout the sampled plates. No extant taxa form a comparable dentigerous ossification embedded in soft tissue that could be analyzed to infer tooth replacement in these plates.

Figure 6 Results of the literature survey on the occurrence of palatal plates in temnospondyl amphibians.

Phylogeny is modified from Schoch (2013). Red lines and asterisks indicate a documented occurrence of the plates; corresponding references are listed in Table 1.

Table 1 Listing of documented occurrences of denticulate palatal plates recovered from the literature review conducted as a part of this study.

All taxa with a superscript number are tentative observations that were not included in the phylogeny and are briefly described here. (1) The authors only identified a shelf similar to taxa in which palatal plates are known; (2) a relatively large denticulate plate on the left ectopterygoid is present, but it is not clear whether this is the natural position; (3) several denticulate plates are noted between the mandible and the displaced clavicle; Schoch (2008) suggested them to be branchial plates but also noted them to be unusually wide; (4) the authors identified isolated denticulate plates in the interpterygoid region but could not determine whether they covered the vacuities or the parasphenoid; (5) a number of ‘scales’ were found in the intermandibular region near the ramus and the cultriform process, but most were removed during preparation; (6) this observation is considered tentative only because the authors were unable to access the original publication; (7) a few small plates with ‘dots’ that probably represent broken denticles are found on the parasphenoid; (8) a large number of variably sized plates in the anterior portion of the vacuity were identified as scleral plates; more plates in the posterior half, as well as denticles, may have been removed during preparation based on photos of the specimen; (9) a number of small denticulate plates are cemented near the base of the cultriform process and were paralleled to those of Chenoprosopus; (10) two isolated patches of denticles are found on the parasphenoid that were suggested to be fragments of the overlying sheet that would have covered the basal plate.

Taxon	Reference	
Acanthostomatops vorax	Witzmann & Schoch (2006:369, fig. 4)	
Adamanterpeton ohioensis	Milner & Sequeira (1998:278, fig. 2)	
Archegosaurus decheni	Witzmann (2006a:148, figs. 8, 17)	
Aspidosaurus binasser	Berman & Lucas (2003:250, fig. 3)	
Balanerpeton woodi	Milner & Sequeira (1993:339)	
“Broiliellus” hektotopos	Berman & Berman (1975:72–73, fig. 2)	
Cacops morrisi	Reisz, Schoch & Anderson (2009:793, fig. 3)	
Colosteid-like tetrapod	Clack et al. (2012:22, fig. 2A)	
Denderpeton acadianum	Godfrey, Fiorillo & Carroll (1987:800, fig. 1D)	
Dissorophus sp.	BMG, pers. obs., 2017	
Eoscopus locklardi	Daly (1994:8)	
Erpetosaurus radiatus	Romer (1930:110, fig. 15)	
Fedexia striegeli	Berman et al. (2010:309; fig. 9B)	
Kamacops acervalis	Gubin (1980:83–88)	
Metoposaurus krasiejowensis	Sulej (2007:56–60, fig. 2E)	
Onchiodon labyrinthicus	Witzmann (2005:481)	
Pasawioops mayi	Fröbisch & Reisz (2008:1020, fig. 3)	
Phonerpeton pricei	Witzmann & Werneburg (2017:1247, fig. 4B)	
Platyrhinops lyelli	Carroll (1964:231–233, fig. 21)	
	Clack & Milner (2010:288, figs. 2c, 6a, c, 7a–b)	
Prionosuchus plummeri	Cox & Hutchinson (1991:568)	
Sclerocephalus haeuseri	Boy (1988:116, abb. 4C)	
	Schoch & Witzmann (2009:148, figs. 6B, E)	
	Witzmann & Werneburg (2017:1247, fig. 4A)	
Sclerothorax hypselonotus	Schoch et al. (2007:122, fig. 3C)	
Siderops kuehli	Warren & Hutchinson (1983:18)	
Stegops divaricata	Steen (1930:862, pl. II, fig. 2)	
	Romer (1930:115, fig. 18)	
Tambachia trogallas	Sumida, Berman & Martens (1998:617, figs. 6–7)	
Trematolestes hagdorni	Schoch (2006:34, figs. 2C, 4B)	
Uranocentrodon senekalensis	Van Hoepen (1915:134)	
1Australerpeton cosgriffii	Eltink et al. (2016:848)	
2Bothriceps australis	Warren, Rozefelds & Bull (2011:743, fig. 3)	
3Callistomordax kugleri	Schoch (2008:91)	
4Capetus palustris	Sequeira & Milner (1993:670)	
5Chenoprosopus lewisii	Hook (1993:284, fig. 2A)	
6Glyptolepis groenlandica	Jarvik (1972: fig. 30) as referenced in Clack et al. (2012:24)	
7Lyrocephaliscus euri	Mazin & Janvier (1983:19, fig. 1)	
8“Metoposaurus” bakeri	Case (1932:22, pl. 2, fig. 5)	
9Nigerpeton ricqlesi	Steyer et al. (2006:23, fig. 2B)	
10Saharastega moradiensis	Damiani et al. (2006:567, fig. 3B)	

Taxonomic perspective

The plates are of little taxonomic utility, partly because their small size and soft tissue attachment result in poor preservation potential. Previous authors have suggested a widespread occurrence within Temnospondyli (e.g., Clack et al., 2012), which is supported by our literature review (Fig. 6 and Table 1). We have also found that they are more common than is apparent from other works that report the presence of the plates, in which they are usually compared to a semi-random handful of other taxa in which plates have been previously described (as well as being erroneously synonymized with branchial plates). Based on the current list of taxa in which the plates are known (Fig. 6 and Table 1), there is no apparent correlation with adult body size, general ecology (terrestrial vs. semi-aquatic vs. aquatic), or inferred feeding ecology (e.g., insectivorous vs. piscivorous). Although it can be reasonably proposed that the plates were reduced or entirely lost along the trajectory that resulted in an absence of denticles on the palatal bones in more derived temnospondyl groups, it should be noted that the plates are found in the Middle Triassic trematosaurid Trematolestes hagdorni, the Late Triassic metoposaurid Metoposaurus krasiejowensis, and the Early Jurassic brachyopid Siderops kehli; in all three, the palatal bones are confidently devoid of denticles (Warren & Hutchinson, 1983; Schoch, 2006; Sulej, 2007). No denticulate palatal ossifications are reported in early lissamphibians such as the albanerpetontids, Triadobatrachus (Rage & Rocek, 1989), or Karaurus (Ivakhnenko, 1978). If the primary function was an extension of the denticulate palatal bones for a gripping surface, then the reduction of denticles could have led to a reduction of the plates, particularly if they share a developmental trajectory. Alternatively, retention of the plates could have balanced the gradual reduction and eventual loss of the denticles on the palatal bones; this may be the condition of the three aforementioned Mesozoic taxa given the evidence for the absence of denticulate palatal bones in these derived forms. If the plates partially functioned as a semi-rigid bracing mechanism within the vacuity for stress distribution, they may have been retained in some capacity for biomechanical purposes.

Importance of nomenclature

One of the other considerations that resulted from this study is the importance of maintaining consistency in definitions and nomenclature in the literature. As noted in the introduction, the definition of a ‘denticle’ varies widely by taxonomic group, and in the case of Drosophila, does not even refer to comparable feeding structures. This is not terribly problematic as the various groups for which a definition of ‘denticle’ exists are so distantly related as to be unlikely to occur within the same study except in the context of a discussion of nomenclature, as in this paper. At present, we do not suggest eliminating the term as it pertains to early tetrapods in light of the longstanding use of the term, as well as the continued merit of the quantitative definition of Lombard & Bolt (2003). However, if additional studies of denticles in other early tetrapods support our findings that the structures are teeth that are simply reduced in size (which is reasonable to expect at present), the definition should be amended to reflect that they are structurally equivalent to true teeth rather than tooth-like protrusions of a more ambiguous nature.

A more serious consideration that should be noted is that a wide variety of names exist to define the denticulate plates themselves; for example, they have been referred to as “denticulate(d) “skin” (Carroll, 1964:233; Berman & Berman, 1975:72), “dermal plates” (Fröbisch & Reisz, 2008:1020), “dermal platelets” (Clack & Milner, 2010(for 2009):288), “dermal scales” (Godfrey, Fiorillo & Carroll, 1987:800), “palatal platelets” (Clack et al., 2012:21), “palatal plates” (Witzmann, 2006b:10), and “palatal ossicles” (Schoch, 2006:34; Witzmann & Werneburg, 2017:1247). These nomenclatural inconsistencies likely account for the previous lack of recognition of such widespread taxonomic occurrence of palatal plates as we have summarized here. Often, different authors use some of these interchangeable terms to refer to different structures. For example, the term ‘dermal plates,’ which was used by Fröbisch & Reisz (2008:1020) to describe the palatal plates of Pasawioops, was alternatively used to refer to palpebral ossifications of Platyrhinops by Clack & Milner (2010 (for 2009):289). Aside from the more common use of the term ‘platelet’ to refer to coagulating blood compounds, it is also not clear as to what, if any, technical distinction exists between ‘plate’ and ‘platelet,’ and because the former appears to be more common in the literature, it should be emphasized. The term ‘ossicle’ is generically defined as a very small bony element, but it is most often used to more specifically refer to bones of the inner ear of various vertebrates, such as mammals (e.g., Rosowski, 1992) and amphibians (e.g., Sigurdsen, 2008; Maddin & Anderson, 2012). It has also been used to refer to denticulate branchial plates (e.g., Milner, 1982:640; Schoch, 2008:91), dermal ossifications (e.g., Case, 1898:519; Schoch et al., 2007: fig. 6H), ornamented plates on the mandible (e.g., Englehorn, Small & Huttenlocker, 2008:299), and scleral ossifications of both anamniotes (e.g., Olori, 2013:402; Schoch & Sues, 2013:441) and amniotes (e.g., Sidor, 2001:1432). Additionally, the term is used with regard to calcareous components of echinoderm exoskeletons (e.g., Maliva, 1989; Dickson, 2004). Although the term ‘plate’ is fairly generic in use, as with the term ‘ossicle,’ we prefer the use of the former because it implies a more defined morphology than the latter, which is a size-based characterization. Furthermore, the problem with any nomenclature that incorporates the term ‘dermal’ is that it implies a similar formation and development to other bony plates that are embedded in the skin, such as scutes and osteoderms, which are found in the connective tissue of the dermis in a wide variety of extant and extinct amniotes, as well as in the dissorophid temnospondyls. It is unclear whether the membranous covering of the interpterygoid vacuities to which these plates were presumably attached was some form of oral mucosa (termed a ‘buccal mucosa’ by Witzmann & Werneburg (2017)) with a structurally homologous outer epithelial layer and sub-epithelial connective tissue. If the membrane was a mucosa, then the term ‘dermal’ would be more appropriate, but this requires further work to better characterize the soft tissue structures of the membrane. Histologically, the plates are clearly distinct from temnospondyl osteoderms (Witzmann & Soler-Gijón, 2010). Additionally, there is no modern analogue in any extant anamniote tetrapod that could shed light on the formation and development of bony plates in the mouth cavity. Therefore, it is not recommended to utilize the term ‘dermal’, as it could incorrectly imply a parallel or a homology to more definitive dermal plates when there are no presently known shared affinities between the two beyond the implantation in a soft tissue. The term ‘tooth plates,’ although not inaccurate in light of the findings presented here, is already used to refer to the dental structures found in lungfish (e.g., Kemp, 1977), chimaeras (e.g., Ishiyama, Sasagawa & Akai, 1984), and ptyctodontid placoderms (e.g., Ørvig, 1985). Accordingly, we believe that it is most appropriate and informative to refer to them as ‘palatal plates,’ which indicates their position in the palatal region and their general flat profile. The complementary use of the descriptor terms ‘denticulate’ or ‘dentigerous’ is acceptable but also nonessential since the palatal plates are always known to bear at least some denticles.

Conclusion

Here we have provided histological evidence that denticles in a temnospondyl amphibian (in this case, those of palatal plates) are structurally congruent with teeth based on the presence of enamel, dentine, pulp cavity with vascular canals, and associated periodontia. This supports the previous work of Bystrow (1938) regarding the denticles of palatal bones of Benthosuchus and provides a phylogenetic (dissorophoid vs. trematosauroid), temporal (Early Permian vs. Early Triassic), ecological (terrestrial vs. aquatic), and positional (palatal plates vs. palatal bones) bracketing on temnospondyl denticles that have been histologically sampled. Given the similarity in external morphology of denticles on the palatal plates and those on the palatal bones across anamniote tetrapods, it seems reasonable to conclude that this characterization is applicable to at least the denticles of other temnospondyls, but additional work across a broad taxonomic range would be necessary to confirm this. We have also provided the first histological description of denticulate palatal plates in a temnospondyl, which can be characterized by the dominance of primary bone matrix, the absence of a cancellous interior region, the absence of secondary remodelling (via the absence of secondary osteons and erosion cavities), the presence of growth marks, a high abundance of Sharpey’s fibers, and a low abundance of vascular channels. Although previous workers have suggested that these plates were implanted in soft tissue coverings of the interpterygoid vacuities, this was formulated mainly on the basis of their position in well-preserved specimens. The presence of abundant Sharpey’s fibers in the palatal plates provides a strong line of evidence to support their positioning within a connective tissue lamina. However, a great deal of uncertainty still pertains to the denticulate plates. Although we have shown that they occupied the interpterygoid vacuities and likely facilitated the capture and movement of prey items, their developmental trajectory and evolutionary origin remain unresolved. The most significant question that requires additional work in the future is the mechanism of tooth replacement on the plates. Despite the large number of plates that were sectioned, we captured only tentative evidence for replacement by origination within the soft tissue and subsequent ankyloses to the plate through the identification of incompletely ankylosed teeth (Fig. 4), and only in one plate (Fig. 2B) did we capture evidence of a dentine fragment located adjacent to a complete tooth that may represent an older, eroded tooth. Additional considerations, such as different rates of replacement throughout ontogeny or during different seasons, cannot be excluded since it is likely that all of the palatal plates of ROM 76838 belong to the same individual. Future work should be directed toward analyzing palatal plates of other taxa to test for any associated phylogenetic differences in plate morphology (e.g., relative thickness of plates, relative abundance of Sharpey’s fibers), as well as to better inform the ontogeny and development (including mode of replacement) in these poorly known structures.

Supplemental Information

Figure S1 Histological sections of ROM 76838 imaged with a lambda filter illustrating major features of palatal plate anatomy

(A) TS01135; (B) TS01137; (C) TS01140; (D) TS01144. Scale bar = 500 µm.

Click here for additional data file.

Thanks to Diane Scott (University of Toronto Mississauga) for assistance and imaging of the specimens prior to the destructive analysis. Thanks to Aaron LeBlanc (University of Alberta) and Mark MacDougall (University of Toronto Mississauga) for discussions of the material. Thanks to Kevin Seymour (Royal Ontario Museum) for providing specimen numbers. Thanks to William May, Richard Cifelli, and other colleagues at the Sam Noble Museum of Natural History for their ongoing support of our research program on the Dolese fauna. Thanks to the reviewers, Florian Witzmann and Pavel Skutschas, and to the editor, Claudia Marsicano, for prompt and constructive comments that greatly improved this manuscript.

Institutional abbreviations

OMNH Sam Noble Oklahoma Museum of Natural History, Norman, OK, USA

ROM Royal Ontario Museum, Toronto, ON, Canada

Anatomical abbreviations

ab alveolar bone

de dentine

en enamel

lb lamellar bone

ode old dentine

pc pulp cavity

pld plicidentine

spf Sharpey’s fibers

vb lines of von Ebner

vc vascular canal

Additional Information and Declarations

Competing Interests

Author Contributions

Data Availability

The authors declare there are no competing interests.

Bryan M. Gee and Yara Haridy conceived and designed the experiments, performed the experiments, analyzed the data, wrote the paper, prepared figures and/or tables, reviewed drafts of the paper.

Robert R. Reisz conceived and designed the experiments, analyzed the data, contributed reagents/materials/analysis tools, reviewed drafts of the paper.

The following information was supplied regarding data availability:

The raw data is included in the figures and tables in the manuscript, and in Fig. S1.

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
