# Peer review of "Histological characterization of denticulate palatal plates in an Early Permian dissorophoid"

_PeerJ, doi:10.7717/peerj.3727_

## Round 0.1 · original submission · Minor Revisions

Dear Mr Gee

You Ms # 17084 entitled "Histological characterization of denticulate palatal plates in an early Permian dissorophoid" which you submitted to PeerJ has been reviewed by two reviewers and the editor.

Both reviewers are coincident that your contribution is suitable for publication and should be accepted after minor revision. In this context, the reviewers have pointed out many changes concerning misspellings, missing references, and rewording, among others.

Please, review carefully all of them and pay attention particularly to suggestions made by Reviewer #2 regarding Figure 2, the incorrect used of the term "anamnmiote" and his suggestion to improve the paleohistology description of the plates.

After this, I think your Ms would be acceptable for publication.

So, I am requesting that you revise the suggestions mentioned above and resubmit your Ms to PeerJ.

Looking forward to receiving your revision.

Sincerely, Claudia Marsicano

·

Basic reporting

No comment

Experimental design

No comment

Validity of the findings

No comment

Additional comments

This is a sound and important piece of work and I support its publication in PeerJ with minor revisions. The following is a list of suggestions for the authors that probably will improve this manuscript. The page numbers refer to the manuscript text proper (i.e. page 1 of the manuscript is page 5 of the pdf)

- Supplemental Table 1: I would include this table in the main article and not in supplemental information, because this table is not large but important, and this would be more reader-friendly
- I did not find a list of anatomical abbreviations in the text
- Page 2, line 58: “…represents the primitive condition within the order”. Please just write “temnospondyls” rather than “the order” because in phylogenetic systematics categories are obsolete
- Page 2, line 60: “The term denticle is also used in reference to…” In the list that you provide here, you may also mention that the term “denticle” is used for pharyngeal teeth in osteichthyans and basal tetrapods (colosteids and temnospondyls); an overview is given by Witzmann (2013) [Reference: Witzmann, F., 2013. Phylogenetic patterns of character evolution in the hyobranchial apparatus of early tetrapods. Earth and Environmental Science Transactions of the Royal Society of Edinburgh, 104(02), pp.145-167]
- Page 3, line 81: “…the early Triassic trematosaurid Benthosuchus.” Please write “trematosauroid” instead of “trematosaurid”
- Page 3, line 88: you cite Witzmann & Schoch (2005); this paper is usually cited as being from 2006
- Page 3, line 89: “…denticulate branchial ossifications, which are also reported from a number of temnospondyls, such as branchiosaurids…” Branchial denticles (either on bony platelets or single without bony base) occur in a wide range of larval temnospondyls but also in certain aquatic adults (with bony platelets: larval and adult stereospondylomorphs, larval micromelerpetontids, larval and adult dvinosaurians, larval eryopids; isolated without bony platelets: larval/paedomorphic branchiosaurids and larval trematopids). Branchial denticulate platelets also occur among aquatic stem-tetrapods in larval and adult colosteids and among “microsaur” lepospondyls in Microbrachis (Olori 2013) (overview in Witzmann 2013, see above). [Reference: Olori, J. C. 2013: Ontogenetic sequence reconstruction and sequence polymorphism in extinct taxa: an example using early tetrapods (Tetrapoda: Lepospondyli). Paleobiology 39: 400–428]
- Page 3, line 92: Please write Onchiodon rather than Onchiodus
- Page 4, line 94: please delete “often” before “feature” and you may add “at one edge” after “only a single row of teeth”. In branchiosaurids and trematosaurids the branchial denticles are isolated because the bony plates are reduced. You may also cite here the review article by Witzmann (2013, see above)
- Page 4, lines 95-96: “Additionally, the presence of branchial plates is indicative of the retention of gills, either in a larval form or a paedomorphic aquatic adult” This statement is right, but should be supplemented by two points:
(1) The presence of branchial dentition is indicative of gill arches (like in fishes or salamander-larvae) with open gill clefts; it does not necessarily indicate (external or internal) gills (although they might have been present); there are examples of extant urodeles with open gill clefts (albeit no branchial dentition) but without gills, e.g. Amphiuma, and then the gill clefts serve mainly for maintaining an unidirectional flow of water during suction feeding and do not serve for breathing (see Schoch, R.R. & Witzmann, F., 2011. Bystrow’s Paradox–gills, fossils, and the fish‐to‐tetrapod transition. Acta Zoologica, 92(3), pp.251-265)
(2) It is correct that branchial dentition is present in many larvae and paedomorphic aquatic adults of basal tetrapods, but they were present also in many aquatic adults like the stem-tetrapod Greererpeton, or among adult aquatic temnospondyls like dvinosaurians, stem-stereospondyls (like Archegosaurus), plagiosaurids, and many trematosauroids. I would not call these taxa paedomorphic; they possessed adult internal gills, and this was probably a heritage from their fish-like ancestors (Schoch & Witzmann 2011; Witzmann 2013)
- Page 6, lines 172-173: “…correlated with the large interpterygoid vacuities that characterize the clade and that are absent in lepospondyls” Although most stem-amniotes including lepospondyls have a rather closed palate, there are some lepospondyl taxa with large interpterygoid vacuities (convergent to temnospondyls): among nectrideans Ptyonius and Diplocaulus (Bossy & Milner 1998), and some “microsaurs” (Carroll 1998). However, palatal plates have never been found in lepospondyls even with large interpterygoid vacuities [References: Bossy, K. A. & Milner, A. C. 1998. Order Nectridea. In Wellnhofer, P. (ed.) Encyclopedia of Paleoherpetology, vol. 1: Lepospondyli, 73–131. München: Verlag Dr. Friedrich Pfeil. Carroll, R. L. 1998. Order Microsauria. In Wellnhofer, P. (ed.) Encyclopedia of Paleoherpetology, vol. 1: Lepospondyli, 1–72. München: Verlag Dr. Friedrich Pfeil.]
- Page 8, line 229: For sake of clarity, please use “palatal bones“ rather than “palatal elements“
- Page 9, lines 247-248: “The plates bearing the denticles are smooth on the dorsal surface (attachment to the epithelium)…” The plates consisting of bone were certainly not attached to an epithelium. Rather, they were embedded in a dermis-like connective tissue (see below).
- Page 10, line 285: “…the enamel is…tapering towards the attachment site”. Does it reach the base of the tooth?
- Page 11, line 334: Please insert “bone cell” before “lacunae”
- Page 11, line 339: “…are oriented roughly perpendicular…to the body of the plate…” Regarding figures 2d and 2f, the fibers seem to be oblique to me
- Page 12, lines 350-351: “One of the main findings of this study is the characterization of the denticles of the palatal plates of a dissorophoid temnospondyl as being structurally identical to true teeth.” In this respect, it is interesting to consider different definition of a “tooth”. Reif (1982) distinguished between teeth and denticles. According to him, true teeth are characterized by a deeply invaginated dental lamina, at which the teeth develop, whereas denticles (both in the oral cavity and on the body surface in certain groups of fishes) develop more superficially in the mesenchyme close to the boundary between epithelium and mesenchyme. According to this definition, the denticles of the palatal plates are not teeth. However, Huysseune et al. (2009) point to the fact that the presence of a dental lamina is not the prerequisite for a tooth to form. Therefore and because of the structural similarities I agree that denticles are homologous to true teeth. [References: Reif, W.-E. 1982. Evolution of dermal skeleton and dentition in vertebrates. The odontode regulation theory. Evol Biol 15, 287–368. Huysseune, A., Sire, J.Y. and Witten, P.E., 2009. Evolutionary and developmental origins of the vertebrate dentition. Journal of Anatomy, 214(4), pp.465-476.]
- Page 12, line 363: Write “Schultze” instead of “Schultz”
- Page 16, line 477: Please write “kehli” instead of “kuehli”
- Page 18, line 529: “…osteoderms, which are found in the epidermis of a wide variety of extant and extinct amniotes”. Osteoderms (and bony scutes or dermal scales) are not located in the epidermis, but in the connective tissue of the dermis below the epidermis. Bone never forms in the epidermis.
- Page 18, lines 530-532: “… the membranous covering of the interpterygoid vacuities to which these plates were presumably attached was probably not skin in the traditional sense, but rather a form of oral epithelium”. In my opinion, it is very unlikely that the membranous covering of the interpterygoid vacuities consisted only of an epithelium. I think it was a kind of oral mucosa that in extant vertebrates is composed of an outer epithelium, a basal lamina, and the lamina propria, the latter consisting of connective tissue. Most probably the palatal plates were formed within in this layer of connective tissue, and without cartilaginous precursor. This is directly comparable to formation of dermal bone formation within connective tissue of the dermis of the skin. So “dermal bone” in the broader sense would not be wrong but you could call it also “membrane bone”. I agree that “palatal plates” is the most appropriate term.
- Page 18, line 550: please replace “trematosaurid” by “trematosauroid”, see above
- Page 19, line 558: the presence of Sharpey’s fibers indicate that the palatal platelets were embedded deeply within connective tissue – comparable to dermis in skin (see above)
- Caption of Figure 5: “(B) representation of the plates during ventral expansion of the epithelium (e.g., during feeding or respiration)”. Please delete “or respiration”. It has been shown that lissamphibians use eye retractation as an aid in swallowing (Deban & Wake 2000; Levine et al. 2004 [References: Deban, S. M. & Wake, D. B. 2000. Aquatic Feeding in Salamanders. In: Schwenk K, editor. Feeding: Form, Function and Evolution in Tetrapod Vertebrates, pp. 65–94. San Diego: Academic Press. Levine, R.P., Monroy, J.A. and Brainerd, E.L., 2004. Contribution of eye retraction to swallowing performance in the northern leopard frog, Rana pipiens. Journal of experimental biology, 207(8), pp.1361-1368], but a contribution to breathing is purely hypothetical! Apart from swallowing, frogs and salamanders may pull their eyeballs in for protection, e.g., during prey capture (Nishikawa 2000) or while the animals are moving [Reference: Nishikawa, K. C. 2000. Feeding in Frogs. In: Schwenk K, editor. Feeding: form, function and evolution in tetrapod tetrapod vertebrates. San Diego: Academic Press. pp 117–147]
- Figure 6: You may unite Trematosauridae and Metoposauridae by red lines, as you did e.g. for Trematopidae and Dissorophidae

·

Basic reporting

no comment

Experimental design

no comment

Validity of the findings

This paper contains the first histological description of the palatal plates of temnospondyls. The paper is a valuable contribution and I strongly support its publication in the PeerJ (after minor revision).

Additional comments

My most relevant suggestions for the improvement of the MS are listed below:


1. Figures.
Generally, figures are clear and informative, but the most of sections are under normal light (authors wrote that Fig2A,F are sections under polarized light (line 284), but I suspect that at least Fig.2A is the section under normal light). I recommend to add few close-ups of the sections under polarized light (may be with lambda waveplate). This could help to show details of orientation of fibers, changes in a density of the bone matrix etc. Because the MS contains the first histological description of the palatal plates, I recommend to provide more details and more figures to make a good background for future studies. I also recommend to add “abbreviations” to figure captions.

2. “Anamniotes”
Authors used a phrase “early anamniotes” in the MS (line 128, “we are principally interested in the characterization of the small palatal plates and their denticles in early anamniotes, with a focus on a specimen from the well-known Dolese Brothers Limestone Quarry near Richards Spur, Oklahoma”). It is unclear, what authors mean here and why a Permian temnospondyl is an early anamniote? For my opinion, the early anamniotes are Cambrian-Ordovician agnathans, but not Permian tetrapods. Later in the text, (line 171, “We make the taxonomic assignment to the Dissorophoidea on several lines of evidence: (1) among extinct anamniotes, denticulate plates are only known in temnospondyls”). Again, the using the term “anamniots” here leads to misunderstanding, because denticulate plates are common, for example, for primitive dipnoans. Later in the text, (lines 307-309, “It has since been found to be widespread throughout numerous anamniotes (often referred to as ‘labyrinthine infoldings’ in older literature), but is also known in a variety of extinct and extant fishes…”), it is also unclear why the authors contrast anamniotes and fishes (fishes are anamniotes!). To sum up, I recommend to avoid the term “anamniotes” in the MS and to use another more precise term (e.g., “non-amniote tetrapods”).

3. Histological descriptions/features
I recommend to expand a description of a bone component of palatal plates (it takes only one short paragraph now) by adding information about the general organization of the bone (absence/presence of diploe structure with three layers), the absence/presence of vascularization, orientation of vascular canals, the absence/presence of secondary remodelling and secondary bone, osteocyte lacunae shape, thickness of Sharpey’s fibers etc. I do not agree that the plates consist of lamellar bone – according to figures it looks like highly ordered parallel-fibred bone, but not as dense as typical lamellar bone that could be found, for example, in the primary and the secondary osteons and in secondary bone trabeculae).
I also recommend to re-phrase identified histological features in “Conclusion sections” (line 558, “… identified features such as growth marks, vascular channels, and Sharpey’s fibres”) and make them more informative (e.g., the presence of growth marks, the absence of the cancellous middle region (= absence of diploe structure), absence of secondary remodelling with the formation of erosion bays and secondary osteons, low number of vascular canals, high number of Sharpey’s fibres and parallel-fibred bone is a dominant type of primary bone matrix.)

Minor comments:
“Introduction”, lines 60-64, usage of the term “denticles”, I recommend to add information that this term is also used in reference to keratinous tooth-like structures in modern agnathans (e.g. lampreys).
.
“Introduction”, line 108, “sp.” after Dissorophus should be in regular (not in italics).

“Taxonomic perspective”, lines 479-480, I recommend to add information that denticulate palatal ossifications were not reported for early (stem) salamanders (Karaurus).

“Importance of nomenclature”, line 540, I recommend to add “and chimaeras” or “and holocephalans” after “in lungfish”, because the term “tooth plates” is used for dental system of chimaeras too. Note that this term also is used for ptyctodontid placoderms.

I waive all anonymity for my review. The authors are welcome to contact me directly should they have questions or anything they wish to discuss.

Pavel Skutschas
29 June 2017
Saint Petersburg State University

---

## Round 0.2 · accepted · Accept

Dear Mr Gee,

It is a pleasure to accept your Ms # 17084, co-authored with Haridy and Reisz, entitled " Histological characterization of denticulate palatal plates in an early Permian dissorophoid" which you submitted to PeerJ.

Thank you for your fine contribution. We look forward to your future contributions to the Journal.

cheers,

Claudia Marsicano